# Trends in reporting embolic and thrombotic events after COVID-19 vaccination: A retrospective, pharmacovigilance study

Yusuke Kan[1,2], Mizuho Asada[1], Yoshihiro Uesawa[1]*

1 Department of Medical Molecular Informatics, Meiji Pharmaceutical University, Kiyose, Tokyo, Japan,
2 Nanohana Pharmacy, Akishima, Tokyo, Japan

* uesawa@my-pharm.ac.jp

**Data Availability Statement:** All relevant data are within the paper and its Supporting Information files.

## Abstract

With the progression of global vaccination against coronavirus disease 2019 (COVID-19), embolic and thrombotic events (ETEs) following COVID-19 vaccination continue to be reported. To date, most reports on the type of COVID-19 vaccine and ETEs have been based on clinical trials, and other reports include a small number of cases. Further, the relationship between the type of COVID-19 vaccine and ETEs has not been clarified. It is important to elucidate trends in the development of ETEs after vaccination, which is a crucial concern for both prospective patients and healthcare providers. In this retrospective, pharmacovigilance study, we analyzed the Vaccine Adverse Event Reporting System (VAERS) reports from January 1, 2020 to June 18, 2021, and performed signal detection and time-to-onset analysis of adverse events by calculating the reported odds ratio (ROR) to understand ETE trends after COVID-19 vaccination based on the vaccine type. Using VAERS, we could collect data about several ETEs associated with COVID-19 vaccination. Nine adverse events associated with ETEs were reported following the administration of viral vector vaccines. The median time to ETE onset was 6 (interquartile range: 2–17) days for mRNA vaccines and 11 (interquartile range: 4–21) days for viral vector vaccines. This study suggests that VAERS aids in disequilibrium analysis to examine the association between vaccine type and ETEs after COVID-19 vaccination. Additionally, the tendency to develop ETEs and the number of days taken to develop ETEs varied depending on the type of the COVID-19 vaccine. Thus, vaccinators and healthcare providers should consider the primary diseases associated with ETEs while selecting vaccines for administration and carefully monitor patients following vaccination for potential ETEs based on the characteristics of vaccine type-specific onset period.

## Introduction

The two types of coronavirus disease 2019 (COVID-19) vaccines currently used in several countries are mRNA vaccines, such as BNT162b2 (Comirnaty®) and mRNA-1273 (Spikevax®), and adenovirus vector-based vaccines, such as Ad26.COV2.S (Janssen COVID-19

**Funding:** The authors received no specific funding for this work.

**Competing interests:** The authors have declared that no competing interests exist.

Vaccine®) and ChAdOx1 nCoV-19 (Vaxzevriar®). Although vaccination is being promoted worldwide to reduce the spread of COVID-19, thrombosis after viral vector vaccination has been reported [1]. The clinical manifestations of embolic and thrombotic events (ETEs), which include arterial and venous ETEs, vary depending on the site of onset and include cerebral venous thrombosis, pulmonary embolism, and myocardial infarction. Importantly, ETEs may adversely affect prognosis. The incidence of thrombosis with thrombocytopenia following vaccination, also referred to as vaccine-induced immune thrombotic thrombocytopenia (VITT) [2], after the first ChAdOx1 nCoV-19 vaccination has been reported as 15.1 cases per million doses, whereas the incidence is 1.9 cases per million doses after the second vaccination [3]. Thrombosis after COVID-19 vaccination is considered by the European Medicines Agency to be rare [4], but it is an adverse event that can have a substantial impact on life expectancy. In some European countries, such as the United Kingdom and Norway, measures have been taken to restrict the administration of the ChAdOx1 nCov-19 vaccine to certain age groups. In the United States, the Centers for Disease Control and Prevention (CDC) and Food and Drug Administration (FDA) recommended a moratorium on the administration of the Ad26.COV2.S vaccine.

Owing to such adverse events, the use of adenovirus vector-based vaccines was discontinued in Japan until August 23, 2021, and only mRNA vaccines were used. Currently, the administration of an adenovirus vector-based vaccine (ChAdOx1 nCov-19) has resumed, but is limited to individuals aged >40 years. In addition, the Swiss Federal Office of Public Health recommends mRNA vaccination and restricts the availability of adenovirus vector-based vaccines [5].

However, no signs of thrombosis have been reported in the clinical trials of each COVID-19 vaccine [6]. Thrombosis with thrombocytopenia syndrome (TTS) after Ad26.COV2.S vaccination has been reported in patients aged 18–60 years [7, 8]. One study suggested that the development of VITT following the administration of ChAdOx1 nCoV-19 could be attributed to a direct interaction between platelets and the SARS-CoV-2 spike protein produced after viral vector vaccination [9], which suggests that the risk of VITT after vaccination with adenovirus vectors may be higher with ChAdOx1 nCov-19 than with Ad26.COV2. The mechanisms and factors that contribute to the development of ETEs after COVID-19 vaccination are still being investigated. Both venous and arterial systems have been reported as the sites of thrombosis caused by TTS after vaccination, and TTS is characterized by cerebral venous thrombosis, which is more frequent and associated with hemorrhage than normal cerebral venous thrombosis. On the other hand, thrombosis after the administration of mRNA vaccines has also been reported in pharmacovigilance studies, prospective or retrospective analyses of vaccinated populations, and large prospective post-marketing studies [10]. To date, few reports have examined the relationship between the COVID-19 vaccine type and ETEs. A recent study that used a large adverse event reporting database reported that the risk of myocarditis/pericarditis was higher with the administration of mRNA vaccines than with that of adenovirus vector-based vaccines [11]. However, studies on ETEs caused by adenovirus vector-based vaccines are limited to a small number of cases. The currently approved vaccines vary across countries. However, as COVID-19 vaccination progresses, many people who are about to be vaccinated, as well as healthcare providers and caregivers, are likely to be interested in the type of vaccine they receive and in knowing the risk of ETEs after vaccination.

The relationships among drugs, adverse events, and the number of days till the development of a specific adverse event have been examined using adverse drug event reporting databases. Therefore, we hypothesized that a similar approach, using a large vaccine adverse event reporting database, could be followed in our study. This study aimed to analyze trends in the incidence of ETEs associated with specific COVID-19 vaccines based on data obtained from a

large vaccine adverse event reporting database. Considering that VITT is caused by the SARS-CoV-2 spike protein produced after viral vector inoculation, we aimed to identify ETEs attributable to the type of COVID-19 vaccine because the tendency to develop ETEs may differ depending on the type of the administered vaccine. We used the reported odds ratio (ROR) method, one of the methods used in disequilibrium analysis, to assess ETEs based on the COVID-19 vaccine type. Importantly, a case report noted the development of cerebral venous sinus thrombosis (CVST) two weeks after vaccination with the mRNA vaccine BNT162b2 [12]. A systematic review reported the development of VITT within 1 week (4–19 days) after the first dose of a viral vector-based vaccine [13]. Despite the small number of cases, these reports raise the possibility of a difference in the number of days until the development of ETEs depending on the type of vaccine. Therefore, we analyzed the number of days to the onset of ETEs based on the COVID-19 vaccine type using the Vaccine Adverse Event Reporting System (VAERS), which includes data on a large number of cases.

## Materials and methods

### Study design and creating a database

We conducted a retrospective, pharmacovigilance study using data from the VAERS database. VAERS is a nationwide passive surveillance system that monitors all adverse events following vaccination in the United States. The database is open to the public and can be accessed free of charge from the respective website following agreement to the terms of use. Signs and symptoms in VAERS reports are coded using Medical Dictionary for Regulatory Activities (MedDRA) terminology recommended by the International Council for Harmonization of Technical Requirements for Pharmaceuticals for Human Use (ICH). The database consists of three data tables: VAERS DATA, VAERS Symptoms, and VAERS Vaccine. VAERS data reported between January 1, 2020 and June 18, 2021 were retrieved for use in the present study.

The flowchart of data extraction for analysis is presented in Fig 1. We excluded incomplete or inaccurately reported data, such as data on cases with unknown sex and the type of vaccine administered, and duplicate data. Additionally, we excluded data on individuals aged <18 years, given that only individuals aged ≥18 years were eligible for COVID-19 vaccination in the United States as of June 18, 2021, as stated in the VAERS data. The time-to-onset analysis table included only data collected since the beginning of the COVID-19 vaccine clinical trials.

### Definition of embolus and thrombus

MedDRA/J ver24.0 Standard MedDRA Queries (SMQ) and System Organ Class terms were used to identify the ETEs. The term "emboli and thrombi" (SMQ: 20000081) includes "arterial emboli and thrombi" (SMQ: 20000082), "emboli and thrombi of unknown or mixed vessel type" (SMQ: 20000083), and "venous emboli and thrombi" (SMQ: 20000084). Therefore, the definition of "embolus and thrombus" (SMQ: 20000081) encompasses embolic and thrombotic symptoms regardless of the vessel type. In addition, "general disorders and administration site conditions," which are considered as local adverse reactions associated with vaccination, were excluded from the ETEs. Further, "injury, poisoning, and procedural complications," "surgical and medical procedures," "social circumstances," and "product issues," which were unlikely to be directly related to vaccination, were also excluded from the ETEs. Therefore, in this study, ETEs were defined after considering these exclusion criteria (S1 Table).

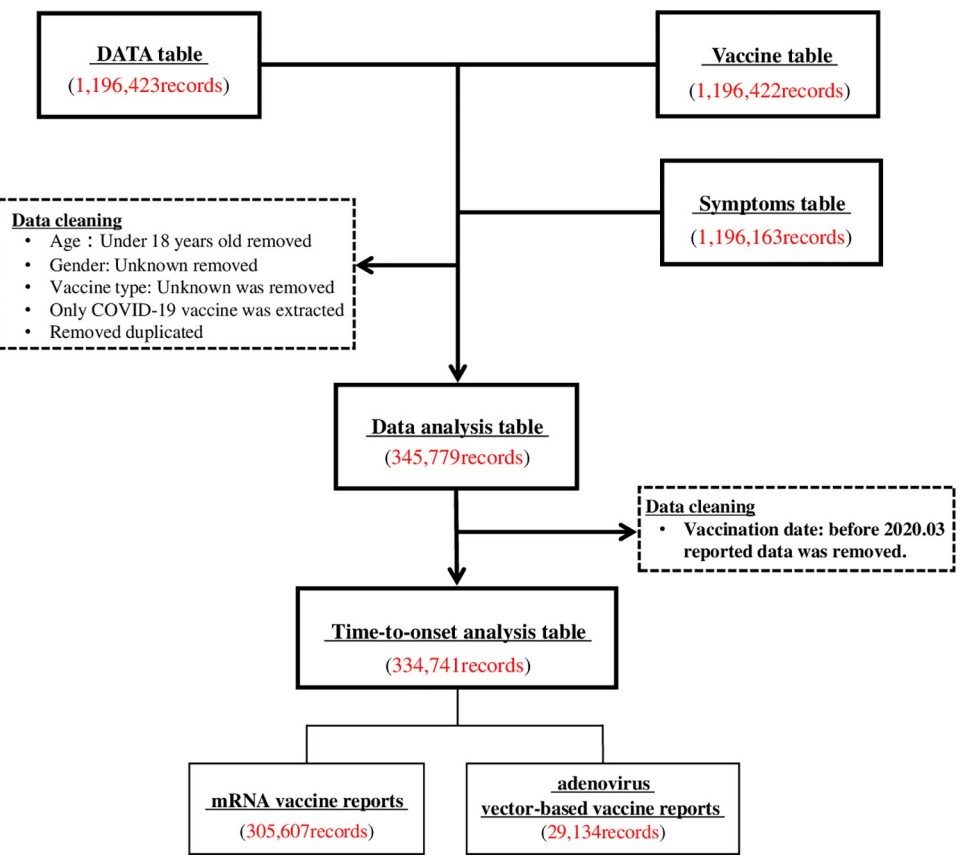

**Fig 1. Flowchart of the database creation.** The VAERS DATA, Vaccine, and Symptoms tables were combined using the patient identification number. Data on cases aged <18 years and on those with unknown sex or unknown vaccine type were excluded, and data on cases that received the COVID-19 vaccine were extracted. Duplicate data were deleted.

### Adverse events owing to COVID-19 vaccination and population

We obtained the demographic data of people who received COVID-19 vaccination in the U.S. from the CDC; we excluded the demographic data of people aged <18 years to account for the age group eligible for COVID-19 vaccination. Using these data and the reports in the VAERS database, we examined the trends in the reporting of adverse events, including ETEs, after COVID-19 vaccination.

### Relationship between COVID-19 vaccine type and ETEs

We extracted adverse events associated with COVID-19 vaccination by detecting signals using disequilibrium analysis in the database created for this study. RORs and 95% confidence intervals (CI) were calculated from a two-row and two-column contingency table (Fig 2), and Fisher's exact test was performed. Because ROR cannot be calculated if there are zero cells in the cross tabulation table, we stabilized the parameter estimation by adding 1/2 to each cell (Haldane-Anscombe 1/2 correction) [14]. Vaccine types affecting ETE incidence were identified by creating a scatter plot (Volcano plot) with the logarithm of the ROR (lnOR) on the horizontal axis and the negative logarithm of the P-value (−log[p]) based on Fisher's exact test on the vertical axis [15, 16]. In this study, adverse events with more than 100 reports were included in the analysis to extract credible RORs based on the precedents of large-scale spontaneous

|  | ETE | other AEs |
|---|---|---|
| adenovirus vector-based vaccine | a | c |
| mRNA vaccine | b | d |

$$ROR = \frac{a/b}{c/d} = \frac{ad}{bc}$$

**Fig 2. Cross tabulations and formulas for reported odds ratios (RORs) of embolic and thrombotic events (ETEs) after coronavirus disease-19 (COVID-19) vaccination.** (a) Number of cases of ETEs caused by viral vector vaccines. (b) Number of cases of ETEs attributed to mRNA vaccines. (c) Number of cases of other adverse events caused by viral vector vaccines. (d) Number of cases of other adverse events attributable to mRNA vaccines. ROR was calculated using the formula shown.

adverse drug event reporting database analyses, such as the FDA Adverse Event Reporting System and the Japanese Adverse Drug Reaction Reporting Database [17, 18].

## Time-to-onset analysis

We used the Weibull distribution for the time-to-onset analysis. The shape parameters of the Weibull distribution can be used to define the hazard without referring to a reference population [17–19]. Cases with blank or unclear onset data and vaccination dates of adverse events were excluded, and cases with an adverse event onset of ≤1 year since the vaccination were included in the analysis. Median onset, quartiles, and Weibull shape parameter (β) were used to evaluate the onset profile of adverse events.

## Statistical analysis

Disequilibrium analysis was performed to detect signals of adverse events. We considered that a signal was present when the lower limit of the 95% CI of the calculated ROR was >1. The Weibull distribution method was used to analyze the time to ETE onset. JMP Pro version 15.0 (SAS) was used for all analyses, and a P-value of <0.05 was considered to indicate statistical significance.

## Results

### Creating a database

Data were extracted from the VAERS DATA (1,196,423 records), Symptoms (1,196,163 records), and Vaccine (1,196,422 records) tables. The final cohort for data analysis included 345,779 records (S2 Table), of which 334,741 records were used for time-to-onset analysis (S3 Table).

### Adverse events due to COVID-19 vaccination

A total of 7144 and 1705 ETEs were reported for mRNA and adenovirus vector-based vaccines, respectively, in the VAERS database. Among the recipients who received at least one dose of

**Table 1. Number of adverse events, including ETEs reported after COVID-19 vaccination and their trends in the United States (January 1, 2020–June 18, 2021).**

| | | Vaccine type | |
|---|---|---|---|
| | Total | mRNA [%] | Adenovirus vector[%] |
| **Total U.S. population (2019) [18+ years]** | 255,369,678 | - | - |
| **People administered at least one dose of COVID-19 vaccine** | 162,637,990 | - | - |
| **Adverse events after COVID-19 vaccination[a]** | 345,779 | 313,123 | 32,656 |
| | | [90.6] | [9.4] |
| **ETEs after COVID-19 vaccination[a]** | 8,849 | 7,144 | 1,705 |
| | | [80.7] | [19.3] |
| **After COVID-19 vaccination[b]** | | | |
| **Number of adverse event cases (per 10,000 population)** | 13.1 | | |
| **Number of ETE cases (per 10,000 population)** | 0.33 | | |
| **After administration of at least one dose of COVID-19 vaccine[c]** | | | |
| **Number of adverse event cases (per 10,000)** | 21.7 | | |
| **Number of ETE cases (per 10,000)** | 0.54 | | |

[a] Number of reported adverse events/ETEs after COVID-19 vaccination in the VAERS database according to vaccine type.

[b] Number of reported adverse events/ETEs after COVID-19 vaccination in the U.S. demographic data according to vaccine type.

[c] Number of reported adverse events/ETEs after COVID-19 vaccination aggregated according to each vaccine type was based on the number of people who received at least one dose of the vaccine and demographic characteristics of the population receiving COVID-19 vaccination.

the COVID-19 vaccine, the number of overall adverse events was 21.7 per 10,000 compared with 0.54 per 10,000 for ETEs (Table 1).

## Relationship between COVID-19 vaccine type and ETEs

Between January 1, 2020 and June 18, 2021, 345,779 individuals were reported to have received COVID-19 vaccinations according to VAERS, of which 8,849 (2.6%) were reported to have thromboembolism. Fig 3 shows the volcano plot displaying the ROR and statistical significance of the relationship between the type of COVID-19 vaccine and the adverse events likely to occur after vaccination. Briefly, adverse events in the upper right corner of the scatter plot are more likely to be caused by viral vector vaccination, whereas adverse events in the upper left corner are more likely to be caused by mRNA vaccination. For the adenovirus vector-based vaccine, the lower limit of the 95% CI of the ROR was >1 for nine ETEs, and a signal was detected (Table 2). In particular, CVST and superficial thrombophlebitis showed a high ROR. The adverse events signaled in this study were mostly consistent with those reported to date. In addition, mRNA vaccines showed a tendency to induce acute myocardial infarction.

## Analysis of ETE-Onset time based on vaccine type

The time-to-onset analysis table included 305,607 records of mRNA vaccines and 29,134 records of adenovirus vector-based vaccines. The median onset times of ETEs after vaccination with the mRNA and adenovirus vector-based vaccines were 6 days (interquartile range: 2–17 days) and 11 days (interquartile range: 4–21 days), respectively. The incidence of ETEs was higher at approximately 1 and 2 weeks after administration of mRNA and adenovirus vector-based vaccines, respectively. The shape parameter (β) of the mRNA vaccines was 0.83 (0.82–0.85), indicating that ETEs tended to develop relatively early after vaccination. For the adenovirus vector-based vaccines, the β value was 1.06 (1.02–1.10), indicating that ETEs tended to occur after a certain period following vaccination (Table 3, Fig 4).

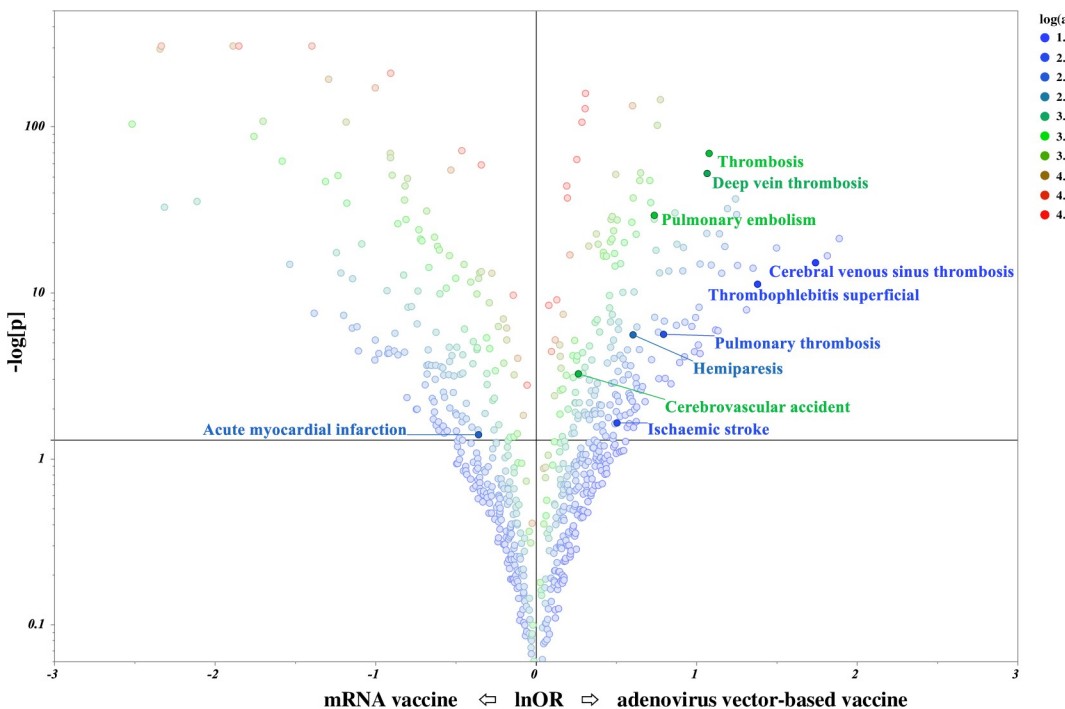

**Fig 3. Volcano plot of ETE according to vaccine type in the United States (January 1, 2020–June 18, 2021).** This figure shows the relationship between the type of COVID-19 vaccine and adverse events. The horizontal axis represents the logarithm of the ROR (lnOR) and the vertical axis represents the negative logarithm of the P-value (−log[p]) determined using Fisher's exact test. The horizontal line in the figure represents the −log[p] = 1.3 (p = 0.05) reference line. The value of a + b indicates the number of adverse events reported, whereas the color change from blue to red indicates the number of reports. The higher the number of adverse events in the upper right corner of the scatter plot, the more likely they are to be induced after vaccination with the viral vector vaccine (S4 Table).

If the shape parameter is or is almost equal to 1 and the 95% CI includes the value 1, the hazard is estimated to be constant over time (contingent failure profile). If the shape parameter is >1 and the 95% CI does not include the value 1, the hazard is estimated to have a maximum value at a specific time (wear-out failure profile).

**Table 2. Reported odds ratios of ETE according to vaccine type in the United States (January 1, 2020–June 18, 2021).**

| Adverse event | ROR | 95% CI | P-value* | a + b[a] |
|---|---|---|---|---|
| Cerebral venous sinus thrombosis | 5.70 | 3.91–8.31 | < .0001 | 111 |
| Superficial thrombophlebitis | 3.97 | 2.80–5.63 | < .0001 | 143 |
| Thrombosis | 2.94 | 2.63–3.27 | < .0001 | 1702 |
| Deep vein thrombosis | 2.90 | 2.56–3.28 | < .0001 | 1318 |
| Pulmonary thrombosis | 2.21 | 1.64–2.99 | < .0001 | 254 |
| Pulmonary embolism | 2.09 | 1.86–2.35 | < .0001 | 1754 |
| Hemiparesis | 1.83 | 1.45–2.31 | < .0001 | 474 |
| Ischemic stroke | 1.65 | 1.12–2.44 | 0.023 | 183 |
| Cerebrovascular accident | 1.30 | 1.13–1.50 | 0.001 | 1587 |
| Acute myocardial infarction | 0.70 | 0.49–1.00 | 0.040 | 419 |

[a] Indicates the number of cases in which the relevant adverse event was reported.

* Fisher's exact test

**Table 3. Median time-to-onset of ETEs after COVID-19 vaccination.**

| Vaccine type | Scale parameter | | Shape parameter | | n | Median | Interquartile range | |
|---|---|---|---|---|---|---|---|---|
| | α | 95% CI | β[a] | 95% CI | | (days) | 25% | 75% |
| mRNA vaccine | 12.2 | 11.8–12.6 | 0.83 | 0.82–0.85 | 6884 | 6 | 2 | 17 |
| Adenovirus vector-based vaccine | 15.4 | 14.6–16.2 | 1.06 | 1.02–1.10 | 1549 | 11 | 4 | 21 |

[a] If the shape parameter and 95% CI are <1, the hazard is estimated to be decreasing rapidly over time (initial failure type profile).

## Discussion

### Adverse events due to COVID-19 vaccination

The total numbers of adverse events and ETEs reported after vaccination with adenovirus vector-based vaccines were lower than those reported after vaccination with mRNA vaccines. However, the proportion of ETEs reported after the administration of adenovirus vector-based vaccines was higher than that of all adverse events. This observation is supported by a considerable number of case reports of ETEs after adenovirus vector-based vaccination.

### Relationship between COVID-19 vaccine type and ETEs

Currently, the mechanism of ETE development after COVID-19 vaccination is unclear. However, VITT and CVST have been reported to be associated with ETEs after viral vector vaccination [1]. The VAERS database includes spontaneous reports; therefore, the direct calculation of risk factors is not possible. However, in this study, we used signal detection index, a commonly used statistical method, to calculate ROR. The nine adverse events that were observed through signal detection in this study did not include VITT but included CVST and thrombosis, especially deep vein thrombosis.

Both the total number of adverse events and the number of ETEs were lower after adenovirus vector-based vaccine administration than those after mRNA vaccine administration. Conversely, the adverse events identified through signal detection were more likely to be induced by adenovirus vector-based vaccines. Adverse events such as thrombosis, including CVST, tended to occur more frequently after adenovirus vector-based vaccine administration than after mRNA vaccine administration; therefore, not only vaccine recipients but also healthcare

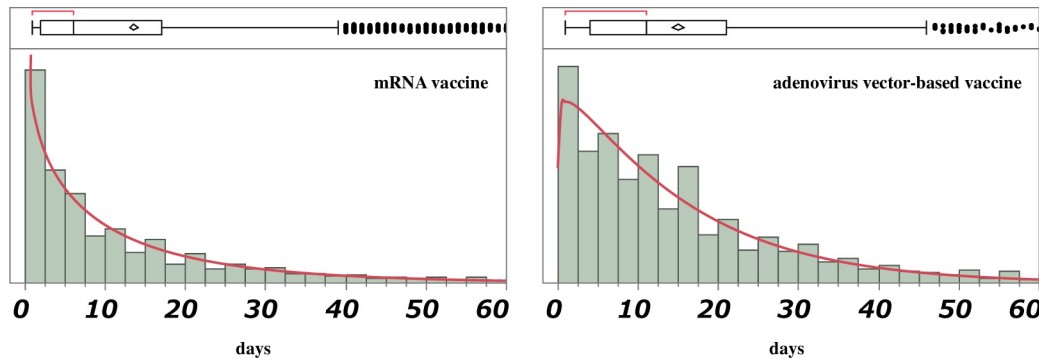

**Fig 4. Histogram of the Weibull distribution in the United States (January 1, 2020–June 18, 2021).** The top of each histogram shows the median (the middle line of the box) and the 25th and 75th quartiles (the two ends of the box). The whiskers represent the farthest point from each end of the box by a factor of ±1.5 and show the largest and smallest values (excluding outliers) of the data within the range. The confidence diamond contains the mean and the 95% CI above and below the mean. The brackets outside the box indicate the shortest range where 50% of the data is dense.

providers must be aware of these adverse events after vaccination. Furthermore, the vaccine type with the least chances of adverse events and ETEs should be considered for administration to recipients with underlying cerebrovascular or cardiovascular disease.

Cases of TTS and CVST have also been reported after vaccination with BNT162b2 and mRNA-1273 [20, 21]. Cases of acute myocardial infarction and myocarditis/pericarditis that were reported as adverse events after the administration of mRNA vaccines are currently under continuous evaluation; however, no significant association between these adverse events and vaccines has been found [22–24]. The present study supports these findings by suggesting that there is a tendency for the development of acute myocardial infarction after mRNA vaccination. Cardiovascular adverse events, such as acute myocardial infarction, are more likely to occur after mRNA vaccination than after adenovirus vector vaccination. As acute myocardial infarction may have a substantial impact on life expectancy, the vaccine type should be considered for vaccine recipients with a current medical history of cardiovascular diseases.

Regarding adverse events related to mRNA vaccines, paroxysmal ventricular arrhythmia related to vaccine administration was reported in subjects aged ≥16 years in a clinical trial of BNT162b2, a COVID-19 vaccine developed by Pfizer and BioNTech [25], and four cases of myocardial infarction were reported in subjects aged ≥18 years in the COVE trial of mRNA-1273, a COVID-19 vaccine developed by Moderna [26]. On the other hand, 11 cases of venous thromboembolism were reported in subjects aged ≥18 years in the EN-SEMBLE study of Ad26.COV2.S, an adenovirus vector-based vaccine developed by Janssen and Johnson & Johnson [27]. However, the small number of reports suggests that their validity as evidence for vaccine-related cardiovascular events is insufficient. We found a significant relationship between vaccines and ETEs based on the statistical analysis of a large-scale adverse event reporting database. We acknowledge that the comparison of RORs in spontaneous reporting databases, such as VAERS, requires careful attention owing to the presence of various biases. To avoid simple comparisons of RORs, we considered the number of reports and the results of Fisher's exact tests together with RORs to achieve semi-quantitative analyses. This analytical approach is based on the hypothesis that a signal detection index supported by the number of reports and P-values has excellent credibility. The current study findings provide further evidence to address the cause for concern regarding the small number of cardiovascular events reported in clinical trials. However, our rationale for a greater risk of acute myocardial infarction with the mRNA vaccines compared with the adenovirus vector-based vaccines is not sufficiently robust because it is based on a borderline P-value (P = 0.04), a relatively mild ROR (ROR = 0.70 [95% CI: 0.49–1.00]), and a relatively small number of reports (n = 419). Clarification of this issue requires the accumulation of data from clinical studies evaluating the risk of vaccine-related acute myocardial infarctions.

## Analysis of ETE-Onset time based on vaccine type

Our analysis of the time to ETE onset based on the COVID-19 vaccine type revealed that the time to ETE onset for mRNA vaccines peaked at 6 days and lasted approximately 2 weeks. For the adenovirus vector-based vaccines, the time to ETE onset peaked at 11 days and lasted approximately 3 weeks, suggesting that the time to ETE onset varies depending on the vaccine type. The findings of the current study utilizing a large database differ from those of clinical reports. However, interpreting the trends in adverse events reported by clinical studies is challenging owing to the small number of reports. Conversely, large databases comprising a large number of reports can reveal trends in ETE-onset time based on the vaccine type as well as identify trends in the occurrence of fatal adverse events, such as ETEs.

## Limitations

Thrombosis after COVID-19 vaccination is a rare adverse event, and the benefits of vaccination must take precedence [28]. The highlight of this study is that individuals yet to receive the vaccine should not decline receiving it, but they should be provided a safety profile after vaccination.

There are several limitations to this study: the VAERS analysis was based on passive surveillance, and reporting bias could have occurred due to both underreporting owing to lack of awareness or compliance with reporting requirements and overreporting owing to increased awareness due to media coverage, among others [29]. Therefore, it was not possible to quantify the true risk. For example, the ROR only examines the increased risk of adverse event reporting and cannot assess the risk of developing adverse events. Therefore, caution should be exercised in interpreting the results from the VAERS database. Further studies including stratification by age and sex are expected to provide more rigorous warnings of adverse events.

## Conclusions

The use of a large vaccine adverse event database was an effective method to collect data on adverse events caused by vaccines that are rarely reported in clinical practice and to evaluate the trend and time of ETE occurrence based on the COVID-19 vaccine type. This study's methodology may complement a small number of post-vaccine adverse event reports, such as those from clinical trials, in post-vaccine adverse events that have a significant impact on patient quality of life. Our study's findings suggest that not only should the population be vaccinated but also that the healthcare workers and caregivers should consider the current medical history of the vaccine recipient during the selection of the specific COVID-19 vaccine type. In addition, as it was suggested that the onset time of ETEs differed depending on the vaccine type, the vaccine recipients should be carefully monitored not only immediately after vaccination but also for a certain period thereafter. Further investigation, including the evaluation of ETEs stratified by age and sex, will provide more details on ETEs associated with COVID-19 vaccination. It is hoped that this will aid in taking precautions against ETEs associated with COVID-19 vaccination.

## Supporting information

**S1 Table. Definition of ETEs.**
(XLSX)

**S2 Table. Data analysis table.**
(CSV)

**S3 Table. Time-to-onset analysis table.**
(CSV)

**S4 Table. Volcano plot.**
(CSV)

## Acknowledgments

We are grateful to the anonymous referees for their constructive reviews.

## Author Contributions

**Conceptualization:** Yoshihiro Uesawa.

**Data curation:** Yusuke Kan, Mizuho Asada, Yoshihiro Uesawa.

**Formal analysis:** Yusuke Kan, Mizuho Asada, Yoshihiro Uesawa.

**Funding acquisition:** Yoshihiro Uesawa.

**Investigation:** Yusuke Kan, Mizuho Asada, Yoshihiro Uesawa.

**Methodology:** Yusuke Kan, Yoshihiro Uesawa.

**Project administration:** Yoshihiro Uesawa.

**Resources:** Mizuho Asada, Yoshihiro Uesawa.

**Software:** Yusuke Kan, Yoshihiro Uesawa.

**Supervision:** Yoshihiro Uesawa.

**Validation:** Yusuke Kan, Mizuho Asada, Yoshihiro Uesawa.

**Visualization:** Yusuke Kan, Yoshihiro Uesawa.

**Writing – original draft:** Yusuke Kan.

**Writing – review & editing:** Yusuke Kan, Mizuho Asada, Yoshihiro Uesawa.

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
