## [Decision Letter · Decision Letter 0]

7 Jan 2022

PONE-D-21-38996Relationship between Vaccine Type and Embolic and Thrombotic Events after COVID-19 VaccinationPLOS ONE

Dear Dr. Uesawa,

Thank you for submitting your manuscript to PLOS ONE. After careful consideration, we feel that it has merit but does not fully meet PLOS ONE’s publication criteria as it currently stands. Therefore, we invite you to submit a revised version of the manuscript that addresses the points raised during the review process.

ACADEMIC EDITOR:

There are several grammar errors and incorrect sentences. Please carefully proofread the manuscript.

We look forward to receiving your revised manuscript.

Kind regards,

Mohamed Hammad, Ph.D.

Academic Editor

PLOS ONE

Journal Requirements:

Additional Editor Comments:

Please be careful to revise your manuscript based on reviewers’ feedback.

Reviewers' comments:

Reviewer's Responses to Questions

**Comments to the Author**

1. Is the manuscript technically sound, and do the data support the conclusions?

Reviewer #1: Partly

Reviewer #2: Partly

2. Has the statistical analysis been performed appropriately and rigorously? 

Reviewer #1: No

Reviewer #2: No

3. Have the authors made all data underlying the findings in their manuscript fully available?

Reviewer #1: Yes

Reviewer #2: Yes

4. Is the manuscript presented in an intelligible fashion and written in standard English?

Reviewer #1: Yes

Reviewer #2: No

5. Review Comments to the Author

Reviewer #1: 1-Please indicate the study design with acommonly used term in the title or the abstract.

2-Please state specific objectives including any prespecified hypothesis ( more details are needed)

In the inroduction last section.

3-in methods section : please present study design early in the paper,describe the setting of the study

In line 65 what is meant by nationwide in which country?

In line 66 what ICH stands for?

State the commercial names of the vaccines mentioned in the study.

For participants state clearly the inclusion and exclusion criteria in obvious point and not to be repeated in many sites in the manuscript and the case definitions as mentioned.

In lines 103&104 what meant by FEARS AND JADER.

STUDY SIZE: explain how it was arrived and the final size.

BIAS:describe efforts to address potential bias.

Statistical analysis: describe all statistical methods used to control for confounders and methods used to examine subgroups and interactions.

4-funding : give the source of funding &the role of funders for the present study.

Reviewer #2: There are clear weaknesses in the paper that the authors must particularly pay attention and handle:

• The major problem of this work is that its novelty and the theoretical contribution are so limited. So, the authors should modify it carefully and improve the novelty of this paper. Also, the authors should provide solid motivation for their work based on the existing literature.

• The abstract must summarize the performance evaluation results.

• The related work papers are only descriptive (1 or 2 sentences per paper) and there are insufficient descriptions of the pros and cons of the work that is cited.

• Figures need to be amended, where the resolution is not clear which makes it difficult to read.

• The results should be further analyzed, more details and further discussion of the simulation results is needed.

• The conclusions section should conclude that you have achieved from the study, contributions of the study to academics and practices, and recommendations of future works.

• The list of references should be reformatted and checked again to be matched with the journal requirement where a different styles and types are used. Please check some spells and typos.

• The paper is hard to read due to the language. The authors should make their manuscript proofread by a native English speaker (lot of typos are avoidable using a speller).

6. PLOS authors have the option to publish the peer review history of their article (what does this mean?). If published, this will include your full peer review and any attached files.

Reviewer #1: No

Reviewer #2: No

---

## [Author Response · Author response to Decision Letter 0]

23 Feb 2022

Response to Reviewer #1

Comment 1

Please indicate the study design with acommonly used term in the title or the abstract.

[Response 1]

We consider the design of this study to be a "retrospective, pharmacovigilance study".

The title has been revised as follows. (P1, L1)

“Relationship between vaccine type and embolic and thrombotic events after COVID-19 vaccination: a retrospective, pharmacovigilance study”

The abstract has been revised as follows. (P1, L16)

“In this retrospective, pharmacovigilance study, we analyzed the Vaccine Adverse Event Reporting System (VAERS) reports from January 1, 2020 to June 18, 2021 and performed signal detection and time-to-onset analysis of adverse events by calculating the odds ratio to understand ETE trends after COVID-19 vaccination based on the vaccine type.”

Comment 2

Please state specific objectives including any prespecified hypothesis ( more details are needed).

[Response 2]

In this study, we thought that by using a large database of adverse vaccine events, we would be able to analyze the trends in the occurrence of ETEs related to the COVID-19 vaccine, for which the number of cases is small in clinical trial data.

The purpose of this study has been revised in the final section of the introduction as follows. (P5, L77)

“The relationships between drugs and adverse events and the number of days till the development of a specific adverse event have been examined using adverse drug event reporting databases. Therefore, we hypothesized that a similar approach, using a large vaccine adverse event reporting database, could be followed in our study. This study aimed to analyze trends in the incidence of ETEs associated with specific COVID-19 vaccines based on data obtained from a large vaccine adverse event reporting database.”

Comment 3-1

in methods section : please present study design early in the paper,describe the setting of the study.

[Response 3-1]

A section "Study design and Creating a Database" was added to Materials and Methods, and the following information about setting up a study design was added. (P6, L96)

In addition, some additions and modifications were made to Fig. 1 to make it more detailed. (Fig1.)

“Study design and Creating a Database

We conducted a retrospective, and pharmacovigilance study using data from the VAERS database.”

Comment 3-2

In line 65 what is meant by nationwide in which country?

[Response 3-2]

VAERS is a nationwide passive surveillance system that monitors all adverse events following vaccination in the United States. 

Therefore, we have made the following amendment. (P6, L98)

“VAERS is a nationwide passive surveillance system that monitors all adverse events following vaccination in the United States.”

Comment 3-3

In line 66 what ICH stands for?

[Response 3-3]

This was the abbreviated name of the International Council for Harmonisation of Technical Requirements for Pharmaceuticals for Human Use (ICH). Therefore, it has been revised as follows. (P6, L102)

“…the International Council for Harmonization of Technical Requirements for Pharmaceuticals for Human Use (ICH).”

Comment 3-4

State the commercial names of the vaccines mentioned in the study.

[Response 3-4]

The name of each vaccine has been added as follows. (P2, L33)

“The two types of COVID-19 vaccines currently used in many countries are mRNA vaccines, such as BNT162b2 (Comirnaty®) and mRNA-1273 (Spikevax®), and adenovirus vector-based vaccines, such as Ad26.COV2.S (Janssen COVID-19 Vaccine®) and ChAdOx1 nCoV-19 (Vaxzevriar®).”

Comment 3-5

For participants state clearly the inclusion and exclusion criteria in obvious point and not to be repeated in many sites in the manuscript and the case definitions as mentioned.

[Response 3-5]

In the definition of ETEs addressed in this paper, System Organ Class terms that are locally or unlikely to be directly related to vaccination have been excluded from this definition.

In the "Definition of Embolus and Thrombus" section of Materials and Methods, we have included the following details regarding the inclusion and exclusion criteria for the definition of adverse events. (P7, L120)

“The term "emboli and thrombi" (SMQ: 20000081) includes "arterial emboli and thrombi" (SMQ: 20000082), "emboli and thrombi of unknown or mixed vessel type" (SMQ: 20000083), and "venous emboli and thrombi" (SMQ: 20000084). Therefore, the definition "embolus and thrombus" (SMQ: 20000081) encompasses embolic and thrombotic symptoms regardless of the vessel type. On the other hand, "general disorders and administration site conditions," which are considered as local adverse reactions associated with vaccination, were excluded from the ETEs. In addition, "injury, poisoning, and procedural complications," "surgical and medical procedures," "social circumstances," and "product issues," which were unlikely to be directly related to vaccination, were excluded from the ETEs.”

Comment 3-6

In lines 103&104 what meant by FEARS AND JADER.

[Response 3-6]

FAERS was an abbreviation for FDA Adverse Event Reporting System and JADER was an abbreviation for the Japanese Adverse Drug Event Report database. We would like to revise the official name. Therefore, we have made the following correction. (P8, L148)

“…, such as the FDA Adverse Event Reporting System and the Japanese Adverse Drug Reaction Reporting Database [17,18].”

Comment 3-7

STUDY SIZE: explain how it was arrived and the final size.

[Response 3-7]

In the Materials and Methods section, in the "Study design and Creating a Database" section, we have added details about cleaning data tables as follows. (P6, 106)

 “Below is the flowchart showing the extraction of data for analysis in the present study (Fig 1). We excluded incomplete or inaccurately reported data, such as data on cases with unknown sex and type of vaccine administered, and duplicate data. Additionally, we excluded data on individuals <18 years of age, considering the age for COVID-19 vaccination in the United States as of June 18, 2021, as stated in the VAERS data. The time-to-onset analysis table included only data collected since the beginning of the COVID-19 vaccine clinical trials.”

We have added a "Creating a Database" section at the beginning of the results and added the scale of the tables created in this study as follows. (P10, L171)

“Creating a Database

Data were extracted from the VAERS DATA (1,196,423 records), Symptoms (1,196,163 records), and Vaccine (1,196,422 records) tables. The final cohort for data analysis included 345,779 records, of which 334,741 records were used for time-to-onset analysis.”

Comment 3-8

BIAS: describe efforts to address potential bias.

[Response 3-8]

As a way to deal with reporting bias in the database, we have set up data table cleaning and definition of adverse events (ETEs). 

For more information on cleaning data tables, please refer to Comments 3-7. 

For the definition of ETEs, please refer to Comment 3-5.

Comment 3-9

Statistical analysis: describe all statistical methods used to control for confounders and methods used to examine subgroups and interactions.

[Response 3-9]

The statistical analysis method used in this study has been added. (P9, L165)

“Disequilibrium analysis was performed to detect signals of adverse events. We considered a signal to be present when the lower limit of the 95% CI of the calculated ROR was greater than 1. A Weibull distribution was performed to analyze the time to ETE onset. JMP pro version 15.0 (SAS) was used for all analyses, and P-value <0.05 was considered significant.”

Comment 4

funding : give the source of funding &the role of funders for the present study.

[Response 4]

No funding was received for this study. The following has been added to clarify the role of the authors. (P21, L354)

“Funding acquisition: The author(s) received no specific funding for this work.”

 

Response to Reviewer #2

Comment 1

There are clear weaknesses in the paper that the authors must particularly pay attention and handle: The major problem of this work is that its novelty and the theoretical contribution are so limited. So, the authors should modify it carefully and improve the novelty of this paper. Also, the authors should provide solid motivation for their work based on the existing literature.

[Response 1]

We have re-examined this paper in terms of its novelty and theoretical contribution.

Novelty: It provides an analysis of the onset time after vaccination using a large-scale adverse event database.

Theoretical contribution: We proposed a vaccine-type selection method for COVID-19 vaccine-associated ETE that takes into account the expression trend and the primary disease and showed the importance of follow-up that takes into account the characteristics of the onset time.

Motivation for this study: Although the expression trends of ETEs reported so far have been obtained from a small number of reports, we thought that a large-scale database of adverse vaccine events would be useful to obtain a more accurate picture of the incidence trends of ETEs.

Comment 2

The abstract must summarize the performance evaluation results.

[Response 2]

The content of the abstract has been re-examined and revised to show the results of the performance evaluation. (P1, L20)

The content of the abstract has been re-examined and revised to show the performance results. 

“Using VAERS, we were able to collect information on a large number of ETEs associated with COVID-19 vaccination.”

(P1, L24)

“This study suggests that VAERS may aid in a disequilibrium analysis to examine the association between vaccine type and ETEs after COVID-19 vaccination.”

Comment 3

The related work papers are only descriptive (1 or 2 sentences per paper) and there are insufficient descriptions of the pros and cons of the work that is cited.

[Response 3]

Thank you for your suggestion.

We have reviewed the research design and the strengths and weaknesses of the cited papers and revised the text. 

In these cited papers, we have added the names of the vaccines and the populations of the vaccine recipients to indicate that there are restrictions on the target age groups and that there have been reports of ETEs in adenovirus vector-based vaccines. (P3, L57)

“Thrombosis with thrombocytopenia syndrome (TTS) after Ad26.COV2.S vaccination has been reported in patients aged 18–60 years [7-8]. One study suggested that the development of VITT following the administration of ChAdOx1 nCoV-19 might be attributable to a direct interaction between platelets and the SARS-CoV-2 spike protein produced after viral vector vaccination [9].”

In this section, we have described the research designs and types of studies reported in the cited references and indicated the shortcomings of the small number of reports. (P4, L66)

“On the other hand, post-vaccination thrombosis has also been reported following the administration of mRNA vaccines in pharmacovigilance reports, prospective or retrospective analyses of vaccinated populations, and large prospective post-marketing studies [10].”

In these cited papers, the names of the vaccines administered were clearly indicated, and the shortcomings of both mRNA vaccines were shown, but the causal relationship is not clear at this point. (P17, L269)

“Cases of TTS and CVT have also been reported after vaccination with BNT162b2 and mRNA-1273 [20,21]. On the other hand, cases of acute myocardial infarction and myocarditis/pericarditis that were reported as adverse events after the administration of mRNA vaccines are currently under continuous evaluation; however, no significant association between these adverse events and vaccines has been found [22-24].”

Comment 4

Figures need to be amended, where the resolution is not clear which makes it difficult to read.

[Response 4]

Thank you for your suggestion. 

We have improved the resolution of the figures. (Fig1-4)

Comment 5

The results should be further analyzed, more details and further discussion of the simulation results is needed.

[Response 5]

In the Discussion section, we have re-examined the results obtained. We have created a section based on the results.

"Adverse Events and Population due to COVID-19 Vaccination" section were added to discuss trends in reporting of adverse events and ETEs after COVID-19 vaccination. Trends in reporting of adverse events and ETEs after COVID-19 vaccination was discussed. (P16, L246)

“Adverse Events due to COVID-19 Vaccination

The total number of adverse events and the number of ETEs reported after vaccination with adenovirus vector-based vaccines were lower than those reported after vaccination with mRNA vaccines. Nevertheless, a higher proportion of ETEs were reported following the administration of adenovirus vector-based vaccines compared with all adverse events. This observation is supported by a noticeable number of case reports of ETEs after adenovirus vector-based vaccination.”

In the "Relationship between COVID-19 Vaccine Type and ETE" section, we reviewed the results on ETE and vaccine type with previous reports and reexamined the suggestions for practice. (P17, L263)

“Adverse events such as thrombosis, including CVT, tended to occur more frequently after the administration of adenovirus vector-based vaccines than mRNA vaccines; therefore, not only recipients but healthcare providers must also be aware of these adverse events after vaccination. Furthermore, the vaccine type with the least chances of adverse events and ETEs should be considered for administration to recipients with underlying cerebrovascular or cardiovascular disease.” 

(P17, L273)

“The present study supports these reports by suggesting that there might be a tendency for the development of acute myocardial infarction after mRNA vaccination. Cardiovascular adverse events, such as acute myocardial infarction, may be more likely to occur after mRNA vaccination than after vaccination with adenovirus vectors. As acute myocardial infarction may have a significant impact on life expectancy, the vaccine type should be considered for vaccine recipients with a current medical history of cardiovascular diseases.”

The section "Analysis of ETE Onset Time by Vaccine Type" discusses the difference between the ETE onset time obtained from the big data and the ETE onset time of a small number of cases reported so far, and the usefulness of a large database. (P19, L305)

“The findings of the current study utilizing a large database differ from those of clinical reports. However, the interpretation of the trends in adverse events reported by clinical studies is challenging owing to the small number of reports. On the other hand, the large number of reports from large databases can uncover trends in ETE-onset time based on vaccine type, and a large database may be useful in identifying trends in the occurrence of fatal adverse events, such as ETEs.”

Comment 6

The conclusions section should conclude that you have achieved from the study, contributions of the study to academics and practices, and recommendations of future works.

[Response 6]

A new section on Conclusion has been added to consolidate research results and recommendations for practice and future research. (P20, L325)

“Conclusions

The use of a large vaccine adverse event database was an effective method to collect information on adverse events caused by vaccines that are rarely reported in clinical practice and to evaluate the trend and time of ETE occurrence based on the COVID-19 vaccine type. Our findings suggest that not only should the population be vaccinated but also that the healthcare workers and caregivers should consider the current medical history of the vaccine recipient during the selection of the specific COVID-19 vaccine type. In addition, the vaccine recipients should be carefully monitored not only immediately after vaccination but also for a certain period thereafter. Further investigation, including studies evaluating adverse events based on stratification by age and sex, might lead to the development of more rigorous warnings about adverse events associated with vaccination against COVID-19.”

Comment 7

The list of references should be reformatted and checked again to be matched with the journal requirement where different styles and types are used. Please check some spells and typos.

[Response 7]

Thank you for your suggestion.

The style of the references has been unified into the Vancouver style.

We also conducted a spell check.

Comment 8

The paper is hard to read due to the language. The authors should make their manuscript proofread by a native English speaker (a lot of typos are avoidable using a speller).

[Response 8]

Thank you for your suggestion.

The manuscript was proofread and spell-checked by a native English speaker.

---

## [Decision Letter · Decision Letter 1]

11 Mar 2022

PONE-D-21-38996R1Relationship between vaccine type and embolic and thrombotic events after COVID-19 vaccination: a retrospective, pharmacovigilance studyPLOS ONE

Dear Dr. Uesawa,

Thank you for submitting your manuscript to PLOS ONE. After careful consideration, we feel that it has merit but does not fully meet PLOS ONE’s publication criteria as it currently stands. Therefore, we invite you to submit a revised version of the manuscript that addresses the points raised during the review process.

We look forward to receiving your revised manuscript.

Kind regards,

Mohamed Hammad, Ph.D.

Academic Editor

PLOS ONE

Journal Requirements:

Reviewers' comments:

Reviewer's Responses to Questions

**Comments to the Author**

1. If the authors have adequately addressed your comments raised in a previous round of review and you feel that this manuscript is now acceptable for publication, you may indicate that here to bypass the “Comments to the Author” section, enter your conflict of interest statement in the “Confidential to Editor” section, and submit your "Accept" recommendation.

Reviewer #1: All comments have been addressed

Reviewer #2: (No Response)

2. Is the manuscript technically sound, and do the data support the conclusions?

Reviewer #1: Yes

Reviewer #2: Yes

3. Has the statistical analysis been performed appropriately and rigorously? 

Reviewer #1: Yes

Reviewer #2: Yes

4. Have the authors made all data underlying the findings in their manuscript fully available?

Reviewer #1: Yes

Reviewer #2: Yes

5. Is the manuscript presented in an intelligible fashion and written in standard English?

Reviewer #1: Yes

Reviewer #2: No

6. Review Comments to the Author

Reviewer #1: As the results of the study were based on ROR and data were self reported this should be obviously indicated in the title to

Avoid misleading so I strongly suggest changing it to start with "Trends in reporting ETE after Covid 19 vaccination ...."instead of the term

relationship .

Reviewer #2: Since the previous version, authors have done some improvement and the paper is much better. Specially there are still main issues that the authors need to handle:

• The abstract must be revised again to include the main performance evaluation results regarding the previous works.

• Please make sure that all keywords have been used in the abstract and the title.

• Figures still need to be amended, where the resolution is so bad which makes it difficult to read.

• The conclusions section still need to revised to address the main contributions of the study to academics and practices, and recommendations of future works.

• A thorough proofreading is still required (best by a native English speaker).

7. PLOS authors have the option to publish the peer review history of their article (what does this mean?). If published, this will include your full peer review and any attached files.

Reviewer #1: No

Reviewer #2: No

---

## [Author Response · Author response to Decision Letter 1]

7 Apr 2022

Response to Reviewer #1

Comment 1

As the results of the study were based on ROR and data were self reported this should be obviously indicated in the title to Avoid misleading so I strongly suggest changing it to start with "Trends in reporting ETE after Covid 19 vaccination ...." instead of the term relationship.

[Response]

Thank you for the suggestion.

The title has been revised as follows to avoid misleading the reader. (P1, L1)

“Trends in reporting embolic and thrombotic events after COVID-19 vaccination: a retrospective, pharmacovigilance study”

Response to Reviewer #2

Comment 1

The abstract must be revised again to include the main performance evaluation results regarding the previous works.

[Response]

This manuscript addresses the fact that previous studies have reported only a small number of cases of ETEs. Therefore, we have revised the beginning of the Abstract as follows: (P1, L12)

“With the progression of global vaccination against coronavirus disease 2019 (COVID-19), embolic and thrombotic events (ETEs) following COVID-19 vaccination have been reported. To date, most reports on the type of COVID-19 vaccine and ETE have been based on clinical trials, and other reports include a small numbers of cases. Further, the relationship between the type of COVID-19 vaccine and ETE has not been clarified.”

Comment 2

Please make sure that all keywords have been used in the abstract and the title.

[Response]

We considered the following keywords in this manuscript: COVID-19 vaccine, Vaccine Adverse Event Reporting System (VAERS), embolic and thrombotic events (ETEs), and reported odds ratio (ROR).

We have added "reported odds ratio (ROR)" to the abstract and confirm that the other keywords are used in the abstract or title. (P1, L18)

“In this retrospective, pharmacovigilance study, we analyzed the Vaccine Adverse Event Reporting System (VAERS) reports from January 1, 2020 to June 18, 2021, and performed signal detection and time-to-onset analysis of adverse events by calculating the reported odds ratio (ROR) to understand ETE trends after COVID-19 vaccination based on the vaccine type.”

Comment 3

Figures still need to be amended, where the resolution is so bad which makes it difficult to read.

[Response]

Thank you for the suggestion.

We have revised Figure 3 to make it easier to read. All figures have been revised to the appropriate size and resolution using PACE (https://pacev2.apexcovantage.com/).

Comment 4

The conclusions section still need to revised to address the main contributions of the study to academics and practices, and recommendations of future works.

[Response]

We have reexamined the Conclusion section and revised it.

The following was added as a contribution to academics. (P20, L330)

“This study's methodology may complement a small number of post-vaccine adverse event reports, such as those from clinical trials, in post-vaccine adverse events that have a significant impact on patient quality of life.”

The following was added as a contribution to practice. (P20, L335)

“In addition, as it was suggested that the onset time of ETEs differed depending on the vaccine type, the vaccine recipients should be carefully monitored not only immediately after vaccination but also for a certain period thereafter.”

We have made the following revisions as recommendations for future research. (P20, L337)

“Further investigation, including the evaluation of ETEs stratified by age and sex, will provide more details on ETEs associated with COVID-19 vaccination. It is hoped that this will aid in taking precautions against ETEs associated with COVID-19 vaccination.”

Comment 5

A thorough proofreading is still required (best by a native English speaker).

[Response]

Thank you for your suggestion.

We have performed the native check again in accordance with your suggestion.

---

## [Decision Letter · Decision Letter 2]

18 May 2022

Trends in reporting embolic and thrombotic events after COVID-19 vaccination: a retrospective, pharmacovigilance study

PONE-D-21-38996R2

Dear Dr. Uesawa,

We’re pleased to inform you that your manuscript has been judged scientifically suitable for publication and will be formally accepted for publication once it meets all outstanding technical requirements.

Kind regards,

Yoshihiro Fukumoto

Academic Editor

PLOS ONE

Additional Editor Comments (optional):

Reviewers' comments:

Reviewer's Responses to Questions

**Comments to the Author**

1. If the authors have adequately addressed your comments raised in a previous round of review and you feel that this manuscript is now acceptable for publication, you may indicate that here to bypass the “Comments to the Author” section, enter your conflict of interest statement in the “Confidential to Editor” section, and submit your "Accept" recommendation.

Reviewer #1: All comments have been addressed

Reviewer #2: All comments have been addressed

2. Is the manuscript technically sound, and do the data support the conclusions?

Reviewer #1: Yes

Reviewer #2: Yes

3. Has the statistical analysis been performed appropriately and rigorously? 

Reviewer #1: Yes

Reviewer #2: Yes

4. Have the authors made all data underlying the findings in their manuscript fully available?

Reviewer #1: Yes

Reviewer #2: Yes

5. Is the manuscript presented in an intelligible fashion and written in standard English?

Reviewer #1: Yes

Reviewer #2: Yes

6. Review Comments to the Author

Reviewer #1: (No Response)

Reviewer #2: The current revision of the manuscript improves the original quality of the original version. Reviewer's comments seem to be satisfactorily addressed. The suggestion is therefore for accepting the paper in the present form.

7. PLOS authors have the option to publish the peer review history of their article (what does this mean?). If published, this will include your full peer review and any attached files.

Reviewer #1: No

Reviewer #2: No

---

## [Editor Report · Acceptance letter]

22 Jul 2022

PONE-D-21-38996R2 

Trends in reporting embolic and thrombotic events after COVID-19 vaccination: a retrospective, pharmacovigilance study 

Dear Dr. Uesawa:

I'm pleased to inform you that your manuscript has been deemed suitable for publication in PLOS ONE. Congratulations! Your manuscript is now with our production department. 

Kind regards, 

on behalf of

Dr. Yoshihiro Fukumoto 

Academic Editor

PLOS ONE